# ‘I Was Present but I Was Absent’: Perceptions and Experiences of the Non-Medical Use of Prescription or over the Counter Medication among Employed South African Women

**DOI:** 10.3390/ijerph19127151

**Published:** 2022-06-10

**Authors:** Nadine Harker, Jodilee Erasmus, Warren Lucas, Diane Deitz, Carrie Brooke-Sumner

**Affiliations:** 1Alcohol, Tobacco and Other Drug Research Unit, South African Medical Research Council, Cape Town 7505, South Africa; jodilee.erasmus@mrc.ac.za (J.E.); warren.lucas@mrc.ac.za (W.L.); carrie.brooke-sumner@mrc.ac.za (C.B.-S.); 2School of Public Health and Family Medicine, University of Cape Town, Cape Town 7700, South Africa; 3The ISA Group, Alexandria, VA 22314, USA; ddeitz@isagroup.com; 4Alan J Flisher Centre for Public Mental Health, University of Cape Town, Cape Town 7700, South Africa

**Keywords:** women, prescription medications, over-the-counter medication, substance use disorders, employment, South Africa

## Abstract

Background: The need for workplace substance use prevention programmes globally and in South Africa is driven by the growing problem of substance use and the associated burden on the health and welfare of employees, their families and organizations. Substance use, which include the non-medical use of medications (both prescription and over-the counter), remains widespread and is a major cause of mortality and a risk factor for non-communicable diseases (NCDs). Method: Twenty in-depth semi-structured qualitative interviews were conducted with employed women in treatment or shortly out of treatment for the non-medically indicated use of over the counter or/and prescription medications (NMIU). These interviews were conducted face-to face with women residing in the Western and Eastern Cape provinces of South Africa. Thematic analysis using NVIVO was used to analyse data collected. Results: The findings from this study suggest that previous use of legal or illegal substances and challenging life experiences underpin pathways to the non-medical use of over-the-counter and prescription medications among employed women. Factors found to contribute to misuse relate to a lack of understanding on risks, and health professional prescribing practices, while mitigators to harmful use were related to increased awareness and understanding harmful practices, the need for improved access and referral to specialist treatment as well as prevention programmes for women. Conclusion: With the improved understanding of the issues surrounding the NMIU of over-the-counter and prescription medications among employed women, the need for interventions to prevent misuse and inadvertently dependency is highlighted.

## 1. Introduction

The non-medically indicated use of over-the counter and prescription medication (NMIU/OTCPRES), which implies using medications for reasons other than those indicated in the prescribing literature or on the box lab [1] and for purposes other than medically intended, has been on the rise since early 2000 and constitute a devastating public health problem both locally in South Africa and globally [2,3]. Over-the counter and prescription medications are easily accessible and while perceived as less harmful when compared to other illicit substances, can lead to the development of substance use disorders (SUDs) and premature death or disability [2]. The World Drug Report of 2019 points to an increase in the misuse of prescription drugs, including opioids, benzodiazepines and synthetic prescription stimulants, stating this to be a growing health problem especially among women in a number of developed and developing countries [4]. Women have been found to be at unique risk for the non-medically indicated use of over-the counter and prescription medications (NMIU/OTCPRES), including a greater burden of dependency and relapse [2,5].

In South Africa, whilst there is still a paucity of data given that the extent and prevalence of NMIU/OTCPRES use is still largely unknown, there has been a growing concern around the non-medically indicated use of medications among women. In 2003, Myers conducted a retrospective study on NMIU/OTCPRES among 9063 patients attending 23 specialist substance abuse treatment centres between 1998 and 2000 in Cape Town. Of the 9063 patients who presented for treatment over this time, 710 (7.8%) were admitted to the NMIU/OTCPRES, or unspecified medicines as either a primary or secondary substance(s) of abuse [6]. This is further corroborated in a more recent publication on women seeking treatment seeking for substance use presenting findings depicting an increase in the proportion of women seeking treatment for medication misuse. The paper further states that when compared to males, patients admitted for codeine use were more likely to be female (*p* < 0.001), when compared to other substances of use. Interestingly, women admitted to specialist substance abuse treatment centres were more likely to be full-time or part-time employed when compared women admitted for other substances of use [7]. Problematic substance use by employed people potentially has negative implications for their health since untreated and undertreated health problems, including harmful substance use, exact substantial costs from individuals and their employers [8]. The need for research into the extent and impact of substance abuse in the South African workforce, and the need to upscale prevention and treatment initiatives through a move away from the traditional labour approach to a more public health approach to addressing substance-related disorders in the workplace, have been underscored [9].

While earlier studies have highlighted risky alcohol use as far more prevalent among employees than problematic substance use [9,10] a retrospective analysis of a large national dataset on Employee Assistance Programme (EAP) service utilisation by an EAP service provider in South Africa which found that while substance use was almost equally distributed across male and female employees, women were significantly more likely to report OTC/PRE-related problems than men [11]. This trend is further substantiated by research in the United States where the problem has been most closely scrutinized. The Centre for Substance Abuse Prevention’s (CSAP) Workplace Managed Care Initiative in the USA indicated that women were much more likely than men to be prescribed abusable prescription medications [12]. Specifically, the data showed that women were significantly more likely to have pain relievers, sedatives, and tranquilizers prescribed to them [13]. This data suggests that women are one of the populations at risk to problems such as an over-reliance on pharmaceuticals, misuse resulting from a lack of adequate information, and vulnerabilities stemming from underlying problems with depression and anxiety [14].

Gender differences regarding motives for the abuse of OTC/PRES medicine have been widely reported in the literature [15]. There are serious associations in between psychological distress and use of prescription medications (opioid and non-opioid) to manage pain, cope with stress and negative emotions in women [16,17,18,19,20]; while men used prescription medications (especially opioids) as a substitute for other (illicit) substances underscored by wating to fit into their social circles [16,20]. This can be seen in a study conducted by Roeloffs et al. (2001) which found that men were more likely than women to use cannabis, cocaine, amphetamines and codeine containing medication [13], while in contrast, women were more likely to (ab)use prescription drugs. Wells et al. (2018) specifically describes codeine-containing over the counter medication as a coping mechanism used to destress from family-life stressors. Additional factors that have been found to contribute to increased prevalence of NMIU/OTCPRES among women include misconceptions these substances are safer to use in comparison to illicit substances, they are more easily accessible from friends and families, that there is greater social acceptability for them by the society and, that women are more likely to be prescribed a variety of substances that have high potential for abuse [14]. While this demographic may be different for certain types of medications and varying drug substitution practices, when considered holistically women appear at higher risk to misuse prescription and non-prescription medications.

Despite the paucity and limitations in available data sources regarding the precise prevalence of NMIU/OTCPRES, it is clear that there remains a public health imperative in South Africa to gain a clearer perspective on non-medically indicated use among women. Given that earlier data suggests that employed women appear to be at a greater risk, consideration should be given to enhancing awareness of NMIU/OTCPRES abuse risk, dependence, and the related harms potentially caused by excessive or long-term use of OTCPRES products in combination. The South African National Drug Master Plan (2019–2024), Goal 3 looks at the Control of Drugs intended for therapeutic use deliverables. Two of these activities involve determining the scheduling status of any substance or medicine based on the risk-access profile of the substance; and to reduce the non-medical use and misuse of drugs and prevent their diversion, misuse, and trafficking. The responsible South African business unit is the South African Health Product Regulatory Authority which impacts local and national policies (NDMP, 2019–2024). In addition to the South African local and national policies, increased research has drawn awareness to the dangers of addictive over-the-counter and/or prescription medication, calling for a need for evidence-based interventions (EBIs).

Prevention efforts to reduce the adverse effects of substance use in various settings including the workplace, coupled with an increased focus on the importance of implementing and disseminating EBIs should be given consideration [9].

The current study aimed to contribute to this knowledge gap, providing a better understanding of the above-mentioned concerns (and contextual factors) surrounding the non-medically indicated use of over-the-counter and prescription medications among employed women. This paper describes perceptions and experiences of the NMIU/OTCPRES among employed women in two provinces in South Africa. This includes the impact of NMIUS/OTCPRES on health and productivity among women who are formally employed, their perceived barriers to accessing help for substance abuse problems, as well as possible approaches for intervention.

## 2. Materials and Methods

This qualitative study is presented in line with COREQ guidelines for qualitative research which included completing a 32-item checklist for each interview [21]. The study followed a qualitative, exploratory, descriptive and contextual design, comprising in-depth semi-structured interviews. Data for this study was collected between March to May 2019.

### 2.1. Population and Sampling

Non-probability purposive sampling informed our sampling framework. The population targeted for this study were either full or part-time employed women who have a history of NMIU/OTCPRES use, residing in the Western or Eastern Cape Provinces of South Africa. These two provinces were selected due to the lead author having established relationships through a past project as well as proximity. The inclusion of 2 provinces as opposed to one was further decided upon to allow for a richer from a wider range of provinces. These women were recruited through specialist substance abuse treatment facilities that provide either in- or out-patient treatment in these regions. Participants were either in treatment at the time of the study or had recently completed treatment for the NMIU/OTCPRES. Our rationale for recruiting women in treatment was to further understand their lived experiences regarding barriers and access to seeking treatment for the abuse/use of NMIU/OTCPRES. Women aged 18–65, who were employed at the time of accessing treatment for NMIU/OTCPRES were included in the study. Participants were excluded if they were not employed full/part-time, under the ages of 18 or older than 65, and did not have a history of OTC/PRES use.

The total sample of women recruited for this study included 20 participants, 10 from treatment centres in the Western Cape Province and 10 from treatment centres in the Eastern Cape Province, respectively. All participants received an incentive voucher valued at ZAR 150 (approximately $10) for their time and participation during the data collection process.

### 2.2. Data Collection and Analysis

Participants were asked to provide written voluntary informed consent after being given a detailed description of the study by research staff. One-on-one semi-structured interviews were then conducted by two experienced qualitative interviewers who were not previously known to the participants using a prepared interview guide (Appendix A). Interviews were conducted in the participants’ preferred language (English or Afrikaans) and lasted between 45–60 min. All participants were fluent in English and preferred the interviews to be conducted in English. In the Eastern Cape, nine interviews took place at a treatment facility, and one telephonic interview took place. However, in the Western Cape, only six of the interviews took place at a treatment facility. Four of the women were snowballed via members of the treatment and recovery community to recruit who had received treatment for NMIU/OTCPRES. Interviews took place in a private room at the at each of the spaces, audio-recorded and transcribed verbatim.

Interview schedules were designed based on the Social Cognitive Theory (Bandura’s Theory) which included the exploration of their outcome expectations, self-efficacy, goal setting and socio-structural variables [22]. Interviews provided an in-depth description of participants’ history with over-the-counter and prescription medication, perceived impacts on their health, social, familial and occupational life, barriers to treatment and perceptions on workplace web-based preventative interventions.

Data were stored and thematic analysis conducted using NVivo 12 software. The thematic analysis was conducted (as a method for identifying, analysing, organizing, describing and reporting emerging themes [23]. Phases of thematic analysis involved six tasks namely; (1) familiarizing yourself with the data; (2) generating initial codes; (3) searching for themes; (4) reviewing themes (5) defining and naming themes; (6) producing the report [24]. Coding the data took place iteratively as information was examined, which led to the noting of emerging themes during the data analysis process. The credibility of the analysis was further enhanced by making use of two independent coders (CBS and JE), and any coding inconsistencies and discrepancies were discussed and agreed upon, to keep codes in alignment.

### 2.3. Reflexivity and Trustworthiness

During data collection, the researchers of this study considered themselves, and the participants included in the study as mutually and continually inclusive, therefore all researchers have acknowledged any preconceived assumptions that may have occurred in relation to the topic under investigation in order to reduce the influence of bias in the study [25]. With respect to trustworthiness, this study adopted Guba’s (1981) elements of quality criteria for naturalistic inquiry to assess the transparency and reliability of qualitative research. The criteria adopted includes (1) dependability, which defines with the stability of the results found over a period of time, (2) confirmability, which directs a great deal of attention on the degree to which findings of an inquiry are a function of the participants and not the biases, motivations and interests of the researcher, (3) transferability, for any findings that may or may not have some type of applicability in similar contexts and participants, and (4) credibility, which is centred around the establishment of confidence in the “truth” of the findings of participants [26].

Additionally, the research team adhered to the following qualitative data collection protocols namely; (1) the use of a semi-structured interview guide for patients included in the study, (2) all interviews were recorded using a dictaphone, and (3) findings of the interviewer were compared to those of the interviewee [27]. To ensure trustworthiness, analysis proceeded to include the triangulation of themes and codes which emerged during data analysis to corroborate findings appropriately. Further to this, researchers utilized the technique of member checking in order to decipher all findings, interpretations and descriptions to ascertain accuracy, and lastly, the researchers team utilized the technique of peer reviewing which entails asking questions and inquiring about the interpretations of the researcher [28].

### 2.4. Ethical Approval

The Human Research Ethics Committee of the South African Medical Research Council provided ethical clearance for this study (Protocol identification number: EC011-7/2018). Written permission was granted by specialist Alcohol and Other Drug (AOD) treatment centres to recruit admitted patients who met eligibility criteria. Further to this, study participants consented to participating in the data collection process independently. Participant confidentiality and anonymity were ensured by using participant codes in transcriptions and to describe research findings. Participation in one-on-one interviews was voluntary and participants were informed that they could withdraw from participation at any stage without repercussion or consequence.

## 3. Results


**Description of participant demographics**


Twenty participants participated in the study with an age range ranged from 20—with a mean age of 42. Years of employment ranged from 6 months to 28 years. In addition, seven of the 20 women held supervisory or management positions (Table 1).

Women in the study described complex life experiences, pathways to substance use and the impact of their use on their physical and mental health, families and work. These are outlined and described in Table 2 as themes and subthemes.


**
*Theme 1: Substances of use reported by participants*
**


Most participants described a history of use of alcohol and other substances from adolescence or early adulthood. The majority described misuse or dependence on more than one substance (prescription, over-the-counter or illicit substances). Commonly used prescription and over-the-counter medications included anxiolytics, stimulants, antidepressants, prescription pain medication, muscle relaxants and sleeping medication. Use of codeine-containing and other over the counter painkillers and cough mixtures were also frequently reported. A minority of participants also reported misuse of illegal substances, cocaine (and crack cocaine), cannabis, ecstasy and crystal methamphetamine (locally known as ‘*tik*’) (Table 3).


**
*Theme 2: Challenging life experiences underpin pathways to use*
**


Participants commonly described a history of challenging life experiences from childhood to adulthood, including difficulty in their current living situations. From these life histories, themes emerged on reasons for use and pathways to use and dependence, as well as the impact of dependence on participants’ lives as outlined below.


*Subtheme 1: Medication use for coping with emotional distress*


Participants described using medications to cope with the emotional impact of loss and grief (e.g., loss of a child), neglect (e.g., by stepparents), trauma (e.g., physical abuse), marital problems (e.g., infidelity), and family conflict. The use of medications to ‘numb’ emotional distress was commonly described. Medications were also used as a coping strategy for stress and anxiety associated with work difficulties, balancing full-time work and parental responsibilities, and improving ability to function with these pressures. Two women described use of ‘pills’ as a learned coping strategy from mothers and grandmothers and a family history of NMIU/OTCPRES dependence.


*‘Especially with the sleeping tablets… I just wanted the pain to go away. I wanted my mind to switch off. I just felt at one stage with the tablets, they helped me to deal, they actually made me dead inside if I can say that. Because I felt at that stage, nothing is working for me, so I have to take these tablets and yes… that and mixed with alcohol isn’t a good thing’.*
Participant 11


*Subtheme 2: Medication use for easing physical issues*


A subset of participants described initially using medications for pain management, including backache, headaches, migraine and post-surgical procedures. Prescriptions to assist with sleep problems were also commonly reported to ‘switch off’, sleep better and have better focus during the day. Two participants reported first using pain medication that was prescribed to a family member, and two described using prescription medications to manage the side effects of their other substance use. One participant also articulated the nuanced link between physical pain and emotional distress and the role medications can play in addressing both.


*‘It was more an emotional thing, you know. So I think it tends to manifest itself physically, your emotions when you get your headaches so on and so forth and you’re trying to numb the pain a little bit. So it’s not just a physical pain that you’re trying to ease, there’s that emotional side of it as well’.*
Participant 5


*Subtheme 3: Medication use for coping in difficult work environments*


Participants reported working in a wide variety of fields, including administration; executive management, retail and restaurant management, entertainment, childcare, health services and several were small-scale entrepreneurs, which is common in the South African work environment. Despite this wide range of spheres of work, participants commonly described the challenging nature of their specific work environments, including conflict in interpersonal relationships, pressure to perform, high workloads and long working hours, all contributing to anxiety and stress.


*For fear of not being productive enough. So for me, my anxiety still is and was around not getting to everything I had to get to every day. So I went through a phase—sort of felt that the Ritalin made me more productive…that was how I felt at the time…then I started to become scared that if I didn’t use that I wouldn’t be able to perform at all.*
Participant 2


**
*Theme 3: Factors promoting misuse of OTC and prescription medication*
**


Most participants reported an underlying mental or neurological disorder, including eating disorders, bipolar mood disorder, post-traumatic stress disorder, epilepsy, general anxiety disorder and depression, and many therefore had experience of prescription of medication by psychiatrists or general practitioners. Despite this, the majority of participants reported their lack of knowledge relating to these medications, and pinpointed their own health professionals as key enablers on their pathway to misuse.


*Subtheme 1: Lack of understanding on risks promotes medication misuse*


The majority of participants described a lack of knowledge related to the medications they used, and they considered this would have been protective in view of the negative impacts their misuse had on their lives. Specifically, they felt more information and discussion with their prescriber would have been important, clarifying the potential for dependency, potential side effects, effects of mixing medications with alcohol, and potential for overdose. One participant expressed the need for prescribers to discuss these issues specifically for medications that are necessary and prescribed for a specific illness (e.g., bipolar) but which nonetheless have potential for dependency and misuse. Most participants also reported not having read medication inserts and one described reading them but not taking medication according to instructions. Some participants felt they were aware of the risks ‘in theory’ but chose to use because of the effect it gave and by the time dependence had developed it was too late to use this knowledge protectively.


*‘But back then it was terrible, I was taking like two, four times a day or something, more than what it says. So I was good at reading the leaflet but I still wouldn’t follow the rules. I would do whatever I thought, which is dangerous’.*
Participant 2


*Subtheme 2: Health professionals are key to misuse*


A minority of participants described the experience of gaining access to medication from friends or colleagues, but the most participants reported that medical practitioners and pharmacists were key to their pathway of misuse. Several participants reported about their long-standing relationships with a medical practitioner or pharmacist through family or friends. Participants reported a range of behaviours for manipulating these practitioners, whilst also highlighting complicity particularly of pharmacists in enabling misuse through negligent dispensing or intentionally taking advantage of their dependence for personal financial gain. Participants commonly mentioned paying cash for prescriptions and regulated OTC medication at pharmacies where pharmacists were known to engage in this behaviour, as well as visiting a number of different doctors and pharmacists. Strategies for manipulating medical professionals included describing and overstating particular symptoms or needs, reporting trauma, family conflict or distress causing anxiety, and using other conditions (e.g., cancer, post-surgery) to justify pain medication. In addition, participants validated their reasons to procure medication from medical professionals by suggesting that that they were ‘visiting’ and had forgotten medication at home. They would also say they needed to retain prescription for use for medical insurance claims (and then re-use the prescription). Alongside this manipulation participants also stated that they had experienced difficulty in trusting medical practitioners and reluctance to disclose dependence on prescription and other medication even when disclosing other dependence (e.g., alcohol). One participant purported changing psychiatrist when their original psychiatrist started questioning their behaviour around medication misuse. Some participants felt they were prescribed too many medications, with a lack of investigation to specific effects of medication and strong doses prescribed by psychiatrists. One participant expressed using illegal substances to counter the side effects of prescribed medication. One of the participants however felt that their treating medical practitioner had a balanced and conservative approach to prescribing.

*‘And my pharmacist, I loved him at the time but I say he should be struck off* (meaning struck off the applicable professional board or council). *Legal dealer, so if I had to see him now, I don’t think I’d be very friendly towards him, not that it’s his fault but how many more people is he feeding drugs to? He is obviously making money for the pharmacy. In fact he also used to give me Tramacet which is a morphine-derivative painkiller. Yes and I got that as often as I wanted’.*Participant 7


*Subtheme 3: Secretive behaviour facilitates escalation of use*


Participants commonly described the escalation of their misuse of prescription medication, along with the awareness that this was ‘taboo’ or socially unacceptable to families, friends and colleagues. Some also reported that they used prescription medication as a more acceptable option, rather than illicit substances as they didn’t want to be seen as a ‘druggie’. This led to secretive behaviour, with participants commonly hiding the extent of use from family and partners, employers, medical professionals and pharmacists, and even within treatment centres. This secretive behaviour was also described for alcohol use by some participants. A minority of participants described extremely high use of medications, despite an awareness of the dramatic effect this would have on their physical health. One reported a suicide attempt using prescription medication.


*‘I would have to take my three (medication for bipolar mood disorder) in the morning to survive. In the afternoon, I’d try to skip one or two. In the evening, I’d try to skip but if I couldn’t then I’d take it which would leave me with additional amounts. But by the time it came to the weekend I’d have sufficient to take between 8 and 16 at a time. I knew I wouldn’t die because I knew I would only die if I drank. So I did a lot of research, if I could die with it. … or, if I was very drunk I would go into the pharmacy and get it myself and keep it at home. And I always kept a very large supply because I didn’t like to ever run out of drugs. So I had 30 s of everything at any given time. And it was easy’.*
Participant 6

Another participant echoes this by saying:


*‘I think the problem is you become so reliant on it that you start justifying more, you justify when you’re lying to the pharmacists, you justify why you’re hiding things, so it just becomes a vicious cycle. The minute you start becoming dependent on something like this all that stuff goes out the window, you lie, you cheat, you steal if you have to, you do whatever you can just to make sure you got that stuff, otherwise you just can’t cope. And I think that’s a big thing that people don’t realise’.*
Participant 3


**
*Theme 4: Perceptions on negative effects of medication misuse*
**



*Subtheme 1: Physical health is compromised*


Although participants acknowledged positive effects of their medication in the short-term including, pain relief, reducing stress, creating a sense of calm and well-being and improving efficiency, all reported negative physical effects associated with extensive use over time. Negative effects included confusion, loss of motor skills, falls and injuries, physical collapse, shaking, memory loss, insomnia, withdrawal symptoms, poor concentration, stomach pain, nausea, panic attacks, loss of appetite and effects of poor nutrition and reduced attention to self-care.


*The memory loss was drastic. I found that’s the hardest to deal with, was my memory loss which I think was the abuse of the sleeping tablets… I had 5 [car] accidents in this last year, so my judgement was completely out from the combination of everything. Although I never experienced a black-out, I’d often not remember the next day, like I forgot to collect my kids from school. I just didn’t deliver my work on time.*
Patient 10


*Subtheme 2: Emotional wellbeing deteriorates*


Despite initial emotional benefits of medications in reducing stress and improving coping, the majority of participants articulated significant emotional distress linked to medication misuse over time. They commonly described disconnection with their emotions, feeling ‘dead’ inside, which fuelled denial of emotional states. Reduced self-esteem related to shame and regret around their addictive behaviour also emerged. Several participants described being in a constant anxious, desperate and overwhelmed state, fuelled by the fear of running out of medications. A minority of participants reported having severe depressive symptoms, suicidal ideation and attempts.


*I hated myself. I was so ashamed I used more…I really just think I wanted to die. I didn’t see a way out. And I didn’t know who I was anymore. The escalation and the self-hate.*
Participant 8


*Subtheme 3: Social interaction and relationships suffer*


Several participants conveyed how their use made them initially more relaxed in social and work situations and increased their self-confidence. However, the majority described the experience and development of social withdrawal and social anxiety as misuse and dependence escalated. Several also reported a reduction in their ability to perform their role effectively at work. The majority emphasized how destructive their misuse was to family relationships. This was manifested as a lack of engagement with family, lack of communication, anger and aggression, leading to conflict with intimate partners and children. One participant reported experiencing sexual dysfunction and several reported separation from intimate partners that they attributed to their dependency. Guilt around behaviour towards children was commonly reported with participants recognizing the harm from being emotionally cut off. One participant described being physically violent towards their child and endangering her children by driving under the influence alcohol and prescription medication.


*‘It was destructive in every area. It destroyed my relationships with my children… it disconnected me from my children. I was present but I was absent. I was too self-absorbed in my own stuff. They felt they couldn’t approach me, it made them fearful of me because I’d become aggressive. I’d become confrontational, manipulative, abusive’.*
Participant 10


**
*Theme 5: Perceptions on how to support women who misuse medication*
**


Some participants described previous experiences of treatment facilities and recovery journeys including periods of abstinence from the use of OTCPRES.


*Subtheme 1: Experiences with access to substance misuse treatment*


Access to treatment was most commonly facilitated by supportive family members particularly intimate partners with help from psychiatrists and psychologists. Self-admission to specialist substance use treatment centres were also reported in some cases, and frequently the cost of in-patient treatment was reported to be predominantly covered by medical insurance, although some participants reported waiting lists at the treatment centres they approached. A minority described judgement and stigma from family members’ and lack of understanding around substance use disorders, and family members with unaddressed dependencies themselves, however most women found no issues in relation to accessing facilities.


*They introduce you to NA and AA. But these are private clinics and I think they’re not readily available for people who can’t afford or they don’t have medical aid. It’s really, really, really scary. There’s not availability.*
Participant 7


*Subtheme 2: Perceptions on priorities for supporting women at risk*


The majority of women expressed the need for women similar to themselves, and others, to know the risks associated with misuse of prescription and over the counter medication. They felt this was a wider scale issue than what would be judged from public perceptions, with many people having some kind of medication ‘in their drawer’. They felt there was a lack of provision of information from doctors, pharmacists, and media channels (e.g., TV, radio, billboards, posters on buses and other transport) which provide information on risks of other substance use. One participant suggested the need for providing this information particularly for girls in schools to promote awareness before the potential for a dependency emerges in their later lives. Participants also commonly described the need for better regulation and accountability for dispensing practices, and registering of clients with misuse. Pharmacists in particular were felt to have a prominent role in providing information and keeping records of patterns of use to highlight when misuse could be emerging.


*You know people are making self-awareness about substance abuse but there is no self-awareness about over the counter medication. That’s the thing; like codeine is like heroin basically. Even Ritalin which is pharmaceutical cocaine, it has the same active ingredient. My brother is actually on Ritalin and he stopped taking it when I came out of rehab and he found it out because he was completely freaked out.*
Participant 19


*Subtheme 3: Perceptions on web-based technology for prevention of medication misuse*


Most participants felt that a web-based approach for information and awareness on behaviour change for medication use would be acceptable and feasible. The primary advantage they reported of using technology in this was to promote anonymity, enabling women to engage with content whilst avoiding the stigma and shame associated with discussing this with medical practitioners or family members. Several participants felt that social media platforms would be more useful for engaging women and some indicated the potential of social media to enable anonymous support groups. Several participants suggested the use of smart phone applications as the most accessible format for technology engagement, suggesting an application which would enable women to input their level of use and assess their risk, as well as finding out more about medications used. Regardless of the technology approach used, participants emphasized the need for having a practitioner to refer women in need, and the potential to access treatment. Participants also described the challenge that whilst this is a practical approach for engaging women in the workplace, they are often overloaded with information from work, junk mail and social media. Any technological approach would therefore need to overcome this barrier to engagement.


*‘I think whoever uses it will be—to be honest I battle a lot with face-to-face discussions, especially at the start of my treatment, whereas if it was online I would have put all the facts right there. It sort of brings safety and anonymity to it.*

*And then I think it is so much more private because people don’t want their families and their friends to know because you’re scared of being judged. People are so judgemental. When you say you’re an addict, they think of a bergie sitting under a bridge, you think of drunks, you think of … But addiction is so much more than that and it is stereotyped. So give people the privacy to log on, maybe make them aware that there is this website that they can log on and how’.*
Participant 2

## 4. Discussion

The non-medical use of prescribed and OTC medicines has climbed steadily over the years [1,29] and is a global concern because of the unpredictable effects of medication misuse especially when used in abnormal/non-prescribed dosages [30]. Whilst much of the data on non-medical use of medications emanate from developed countries, this paper contributes to current research by describing the perceptions and experiences of employed South African women who have used either prescription or over-the-counter medications for non-medical purposes. Major themes emerging from our study refer to the experience of challenging life circumstances which underpin pathways to use and prescribing practices which potentially contribute to the escalation to harmful use. Perceptions of negative aspects of medication use as well as measures to support women also emerged as important findings that will help inform public health interventions that target employed women.

Findings from this study indicate that some participants had used alcohol and/or other substances of use recreationally during adolescence and/or early adulthood. A recent scoping review suggests that early initiation of substance use dramatically increases the risk of and vulnerability to lifelong substance use disorders (SUDs) including alcohol use disorders (AUDs) [31,32,33]. This is further corroborated by Jordan and Anderson (2017) suggesting that individual risk for SUD emerges from an immature prefrontal cortex combined with hyper-reactivity of reward salience, habit, and stress systems. Early initiation or exposure to substances therefore bring changes to the prefrontal cortex activity that can persist into adulthood [31]. This finding is important as it places emphasis on the need for concerted prevention efforts to ensure that young people, especially young girls at risk, are identified early and provided with interventions to help mitigate the use of the same or alternate substances in later years. In line with Badura’s theory, such efforts will also promote self-care behaviours as a protective factor for young girls [22]. Targeted, tailor-made gender specific preventative interventions are therefore needed to maximise resilience to developing a SUD in later life [31].

Challenging life circumstances, particularly the experience of bio-psycho-social and occupational related distress have also been identified as facilitators to the non-medical use of medications. The experience of stress and anxiety associated with either work difficulties, balancing full time work or mothering/parenting responsibilities, underlying mental health issues and improving ability to function with these pressures are some of the reasons participants gravitated to the non-medical use of medicines. We know from published work that the use of, for instance, codeine-containing products as a coping mechanism have been found to be a root cause for misusing medications [29,30]. This can be exacerbated with the existence of underlying mental disorders (e.g., depression) which is also a known risk factor in the development of substance use disorders and relapse vulnerability [2,4,34]. Additionally, health and medical needs of South Africans are complex and compounded by a variety of living conditions, differences in socioeconomic statuses and educational access. Notably, South Africans face a quadruple burden of disease with HIV/AIDS, tuberculosis and non-communicable diseases, as well as violence and injury. It can be argued that these further exacerbate the impact on OTC/PRES dispensing [35]. Whilst emotional and psychological distress was cited as one of the conduits to the non-medical use of medications in this study, a sub-set of participants reported initially using medications for pain management, including backache, headaches, migraine and for post-surgical procedures. Women are particularly at risk for the non-medical use because women have more doctor’s visits, report higher rates of pain, and are more likely to be prescribed pain medication for longer periods of time and at higher doses [4,29]. In contrast, one South African study did report that more women are seeking treatment for alcohol and other drug use problems, however, women do face unique barriers to accessing treatment for substance use disorders, such as grappling with stigma, geographical location to treatment, financial constrictions to access treatment, and responsibilities surrounding childcare, amongst others [7], further motivating the need for specialist care strategies for women to receive treatment and medical guidance without facing these challenges.

In further exploring the drivers or enablers to the non-medical use of medication from the perception of the women who participated in this study it is evident from our findings and corroborated by other literature sources that factors such patient-dispenser and patient-prescriber relationships as well as overall knowledge and awareness play a role in the non-prescribed use of medication. Over-prescribing practices for physical problems such as for pain management as well as for mental health problems are thought to exacerbate the problem requiring more vigilance on the part of the prescriber and the dispensing pharmacist. To remedy this, literature references the importance of “responsible prescribing” which should include risk and screening assessments, prescribing agreements and treatment contracting without compromising legitimate access to medications [32,36]. Pharmacovigilance in prescribing opioid or other habit-forming medications is needed to monitor use not only generally but also in vulnerable groups such as women and young people [32]. It has also been found that any failure on the part of physicians to properly screen, diagnose, and or counsel a patient prior to the initiation of treatment for OTC and/or prescription drugs can be a contributing factor [8]. Additionally, better regulation and accountability for dispensing practices, and registering of clients with misuse should be considered [37].

Also emerging from these findings are women’s perception on knowledge and awareness on the risks associated with the use of these medicines. An absence of knowledge and awareness was thought to be a conduit to misuse, with participants recommending interventions that support women at risk. Eaves (2015) describes individuals who misuse as not being aware of their dependency to over-the-counter medication and calls it ‘engaging in ideological harm reduction’ as individuals avoid making any associations to the addictive potential of OTC or prescription medication. Of further concern is the diversion of medications to the “black-market” which is where OTC/PRES medications are loosely accessible by anyone in South Africa [38]. A local study looking at the prevalence and correlates of prescription drug diversion and misuse among people living with HIV also cited several comparable challenges to those raised in our study interviews. For instance, the authors suggested that those initially prescribed medications such as codeine did not know of their dependency potential and regarded their use as appropriate [39]. Lack of awareness of the addictive potential of sustained medication use, including the risk of dependence and harm [39] were also reported, again echoing findings from our study. Notably, while the recognition of the pleasurable effects of NMIU/OTCPRES medications were acknowledged and perceived helpfulness mentioned, participants highlighted an overall lack of awareness of forms of tolerance and harmful patterns of use although some were aware of the negative impacts. Given the above, it is imperative that the therapeutic relationship between prescriber/dispenser and patient include discussion which seeks to clarify the potential for dependency, potential side effects, effects of mixing medications with alcohol, the dangers of sharing medications and potential for overdose [37]. Moreover, we also found that a lack of awareness is driven by the user’s ability to read and understand the medicine information leaflet, a finding also evident from other studies (4). Medication leaflets are often written in small print and often in convoluted language that may not be understood by the user. This should be mitigated by innovative ideas and strategies for user-friendly leaflets for medications that have an increase potential for misuse [40] as suggested by participants in this study. Additionally, in South Africa not all pharmacies are part of a national register which is often able to track over-prescribing practices and therefore issue an alert.

Experiences shared by participants in this study suggest the need for support for employed women at risk. One approach with potential in this area is the use of interactive web-based prevention programmes as a mechanism for enhancing knowledge and awareness as well as educating women on responsible use of medications. This is in line with the social learning theory which emphasises the importance of unveiling new methods of teaching, information sharing and brining about awareness of a behaviour that is healthy to an individual [22]. Most participants were of the view that a web-based approach for information sharing, education, and awareness on behaviour change for medication use would be acceptable and feasible. Web-based technologies were thought to promote anonymity, and avoid the stigma and shame associated with substance use including the non-medical use of medications. The dawning of new technologies through increased availability of the internet present unique opportunities for addressing an array of health problems, including substance use disorders. Capitalizing on this opportunity to upscale digital technologies (also referred to as e-health) on interventions for the treatment of patients via the internet or computer-based interventions for SUDs, as well advance self-seeking education on SUDs, patient monitoring and referrals to resources and professionals are now attainable due to e-health [41].

Participants of this study were also of the opinion that social media platforms would be useful for engaging women and some indicated the potential social media has to enable anonymous support groups. In the literature, this is referred to as m-health, short for mobile health, whereby health resources are made available through mobile devices such as smartphones and tablets [41]. Several participants suggested the use of smart phone applications as the most accessible format for technology engagement. This finding is particularly encouraging as it has the potential to combat stigma and discrimination which may hinder access to care [41,42] and may contribute to the early identification of young women at risk since women suffer more social stigma, and may avoid help seeking due to fears of stigma towards their families [43]. Interactive web-based interventions offer unique opportunities for disseminating behavioural and other public health education materials [12] with anonymity which may be particularly important for working women, and young women. The advantages of web-based programmes also include its ability to reach women who work in environments where competing demands and time constraints make it difficult to attend traditional programmes or interventions [44] since these platforms can accommodate personal needs, time and interests [45,46]. Additionally, with increased access to digital technologies and mobile devices, the fostering of self-management and monitoring of personal health may provide users with a cost-effective method to access to healthcare information. Furthermore, there is a room in workplaces, as part of standing employee wellness programmes (EWP), for such web-based prevention programmes that target not only women but the entire workforce, again reinforcing the de-stigmatising potential of the approach. There is a large body of evidence suggesting that workplace prevention programmes not only provide information, education and awareness, but also build skills to empower employees to make healthy choices such as improving the willingness to seek help (web-based programmes should be tailored to include online screeners), increase utilisation of health services and benefits, reduce high-risk behaviours such as substance use and reduce employee experience of stress at work [3,11,47,48].

## 5. Conclusions

This study holds particular value for workplaces since the topic of substance use prevention holds appeal to workplaces hoping to reduce health care costs, absenteeism, injuries, staff-turnover further facilitating referral to treatment if so required. With the improved understanding of the issues surrounding non-medically indicated use of over-the-counter and prescription medications among women, consideration could be given to interventions that help prevent harmful use.

## Figures and Tables

**Table 1 ijerph-19-07151-t001:** Description of sample.

Employment Type	Age	Management Position	Years Employed	Marital Status	Dependents
Psychological Career	54	No	9 months	Divorced	0
Psychiatric Recovery Assistant	48	Yes	20 years	Divorced	2
Financial Administrator	39	Yes	6 years	Married	2
Educator	30	No	8 years	Engaged	0
Human Resources	49	Yes	12 years	Single	3
Administration	34	No	11 years	Single	0
Secretary/Office Manager	47	No	28 years	Married	2
Administration	46	No	6 months	Married	2
Au pair	21	No	4 years	Single	0
Bookkeeping	40	No	20 years	Married	2
Administration	22	No	6 months	Single	0
Needlework Facilitator	42	No	2 months	Married	2
Recovery Assistant	42	Yes	25 years	Single	2
Psychiatry	51	Yes	12 years	Single	2
Administration	33	Yes	6 years	Single	0
Telecommunications	35	Yes	10 years	Divorced	2
Entrepreneur in Arts and Crafts	46	No	23 years	Divorced	2
Examinations Administrator	27	No	10 years	Single	0
Administration Assistant	46	No	unknown *	Married	2

* Demographic of one participant missing.

**Table 2 ijerph-19-07151-t002:** Experiences and perceptions of South African women on use of prescription or over-the- counter medication.

Theme	Subthemes
Theme 1: Substances of use by participants	No subthemes
Theme 2: Challenging life experiences that underpin pathways to use	Subthemes: Medication use for coping with emotional distress Medication use for easing physical issues Medication use for coping in difficult work environments
Theme 3: Factors promoting misuse and escalation of use of OTC and prescription medication	Subthemes: Lack of understanding on risks promotes medication misuse Health professionals are key enablers to medication misuse Secretive behaviour facilitates escalation of use
Theme 4: Perceptions on negative effects of medication misuse	Subthemes: Physical health is compromised Deterioration of Emotional wellbeing Social interaction and relationships suffer
Theme 5: Perceptions on how to support women who misuse medication or who are at risk	Subthemes: Experiences with access to substance misuse treatment Perceptions on priorities for supporting women at risk Perceptions on web-based technology for prevention of medication misuse

**Table 3 ijerph-19-07151-t003:** Description of substances used by participants in the study.

OTC/PRE-Medication	Number of Women Using	Medication/Substances Used *
Alcohol	8	Not specified
Illicit Substances	5	Dagga, Crack/Cocaine Methamphetamine
Sleeping Tablets	6	Not specified
Tranquilizers	5	Espiride
Anti-Anxiety	9	Ativan, Rivotril, Xanax, urbanol
Codeine containing medication	2	Genpain, Myprodol, Morphine, Panamol, Adcodol, Mybulen, Cough mixture, allergex, Painstop, Stilpain, adco-sinus, sinu-tab, sinu-max
Medication without codeine	4	Grandpas
Decongestants	1	Ephedrine
Slimming tablets/Diuretics	1	Phedra-cut
Anti-Depressants	7	Wellbutrin, Nuzak
Stimulants	3	Ritalin
Muscle relaxants	2	Voltarens, Nurofens, Norflex

* not all women stipulated specific substances. Column represents only a few of the medications used.

## Data Availability

Data can be made available upon formal request.

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
