# Peer review of "‘I Was Present but I Was Absent’: Perceptions and Experiences of the Non-Medical Use of Prescription or over the Counter Medication among Employed South African Women"

_ijerph, 2022, doi:10.3390/ijerph19127151_

Round 1

Reviewer 1 Report

Review report

Brief summary 

The aim of the paper is to provide a better understanding of the experiences of the NMIU/OTCPRES among employed women in two South African provinces. This paper describes these women’s perceptions and experiences of the NMIU/OTCPRES.

The main contribution of the paper is:

The experience of stress and anxiety associated with either work challenges, balancing full time work or mothering/parenting responsibilities, underlying mental health issues and improving ability to function with these pressures are some of the reasons women gravitated to the non-medical use of medicines.

Over-prescribing practices for physical problems such as for pain management as well as for mental health problems exacerbate the problem requiring more vigilance on the part of the prescriber and the dispensing pharmacist.

Pharmacovigilance in prescribing opioid or other habit-forming medications is needed to monitor use; generally, but more specifically in vulnerable groups such as women and young people.

Strengths of the paper:

The study is relevant in the current milieu of SUD research focusing on women’s wellbeing in SA and globally considering the challenges that employed women are experiencing during the covid pandemic and will be experiencing in a post-covid era.

Advocating for social media platforms for engaging women and to enable anonymous support groups for working women.

The methodology is sound and well-articulated.

The findings are appropriately contrasted and compared to previous studies.

General comments

Article: The qualitative approach and research design were appropriately selected and well executed. Theoretical foundation is also well-articulated in the paper.

Review: The topic is comprehensively covered, with the relevant sources provided to substantiate the need and support for the study. The comparison and contrasting of the literature and subsequent literature control to substantiate the findings is well executed and provides for a linear flow of information and presents a coherent whole of the discussion which could be easily followed. The findings are comprehensive and brought to the fore some emerging ideas in terms of gender specific interventions and the need to advocate for more responsible and monitoring dispensing of NMIU/OTCPRES. Thus, the study therefore addresses the gap in knowledge which is identified in the rationale. Appropriate and a wide assortment of sources were used and correctly referenced in line with the journal requirements.

Other comments:

The paper is clear, relevant for the field and presented in a well-structured manner.

There is a mix of recent and publications older than 5 years.  There are 4 references by the first author which is acceptable, considering the exhaustive list of references.

The paper is scientifically sound and the qualitative approach, the exploratory, descriptive and contextual design is appropriate considering the nature and aim of the study.

The study results are dependable and therefore reproducible based on the detail description given in the methods section of the paper.

The three the tables are appropriate, properly indicating the data. They are easy to interpret and understand. The data is interpreted appropriately and consistently throughout the manuscript.

The conclusions are consistent with the evidence and arguments presented.

The ethics statements are adequate.

In terms of the data availability statement, the authors indicated NA.  A statement is required.

Specific comments 

144

The rationale for the selection of the two provinces is not provided.

19,168,189,209,

496, 656

The interchangeable use of first- and third-person style of writing needs to be addressed.

172

Should read, (Bandura's theory). Theory should be written in sentence or lower case, not uppercase.

173

Close bracket

175, 490

Use acronym, OTCPRES introduced earlier in the paper.

324-346

Condense to make this paragraph shorter, and therefore consistent with the rest of the presentation of the discussion of the findings.

348

Use brackets to specify the context/meaning in the quote: … he should be struck off (Professional Board/Council)?

590

Remove the editorial comment in the manuscript. Add reference to substantiate the statement.

655

Provide a data availability statement.

Rating the Manuscript

·    Novelty: The question is original and well-defined. The results provide an advancement of the current knowledge.

·    Scope: The work fit the journal scope.

·    Significance: The results are interpreted appropriately and are significant. All conclusions are justified and supported by the results.

·    Quality: The article is written in an appropriate way. The data and analyses thereof are presented appropriately.

·    Scientific Soundness: The study was correctly designed and technically sound. The analyses were performed with the highest technical standards. The data is robust enough to draw conclusions. The methods, tools and software are described with sufficient detail to allow another researcher to reproduce the results. The raw data was not made available, but this is not a requirement. However, a statement regarding its availability should be included where applicable in the outline of the paper.

·    Interest to the Readers: The conclusions are interesting for the readership of the journal. The paper has the potential to attract a wide readership.

·    Overall Merit: There is an overall benefit to publishing this work. See previous comments regarding the scope and relevance of the topic. The work will definitely advance the current knowledge, especially relating to gender specific interventions aimed at working women.  

·    English Level: The English language is appropriate and understandable.

Overall Recommendation

·    Accept after Minor Revisions: The paper can in principle be accepted after revision based on the reviewer’s comments. Top of Form

Author Response

Dear Reviewer, thank you for taking the time to review this manuscript.

Please find our responses below.

In terms of the data availability statement, the authors indicated NA.  A statement is required.

       We have included the statement “Data can be made available upon formal request”.

  1. The rationale for the selection of the two provinces is not provided.

We have inserted the following rationale: These two provinces were selected due to the lead author having established relationships through a past project as well as proximity. The inclusion of 2 provinces as op-posed to one was further decided upon to allow for a richer from a wider range of provinces. 

  1. The interchangeable use (19,168,189,209,496, 656) of first- and third-person style of writing needs to be addressed.

Duly corrected.

  1. Should read, (Bandura's theory). Theory should be written in sentence or lower case, not uppercase.

Duly Corrected.

  1. Close bracket (173)

Duly Corrected

  1. Use acronym, OTCPRES introduced earlier in the paper.

Duly Corrected

  1. Condense to make this paragraph (324-346) shorter, and therefore consistent with the rest of the presentation of the discussion of the findings.

We have shortened this paragraph.  

  1. Use brackets to specify the context/meaning in the quote: … he should be struck off (Professional Board/Council)?

Corrected and added the context to the statement.  

  1. Remove the editorial comment (590) in the manuscript. Add reference to substantiate the statement.

We have removed the editorial comment

  1. Provide a data availability statement

A data statement “data can be made available upon request’ was added.

Many thanks for your consideration of the manuscript.

Reviewer 2 Report

Dear Authors, 

I was really pleased while reading your article. Not only you've investigated a very up-to-date topic which might be in the scope of interest for readers worldwide but also you've chosen a valuable and valid methodological approach.

Strengths of the manuscript include among others:

- qualitative, phenomenological approach to investigate respondents' experiences

- study design supported by Bandura's theory

- the use of COREQ protocol for reporting the research

- detailed discussion on study ethics and trustworthiness

To enhance the repeatability of the study in other regions of the world I would suggest adding the interview guide.

It might be also sound to discuss your results a bit more in the context of chosen theory (The theory should not only guide the study protocol but also data analysis and discussion - at least indirectly)

Although there are some editorial flaws (e.g line 20 and 173, 643), these do not affect the quality of the paper and may be corrected at further steps of paper proceeding.

Author Response

Dear Reviewer, Thank you for taking the time to review his manuscript. 

Please find our changes below: 

  1. To enhance the repeatability of the study in other regions of the world I would suggest adding the interview guide.

We are happy to include the interview guide as supplementary material. This is attached. 

  1. It might be also sound to discuss your results a bit more in the context of chosen theory (The theory should not only guide the study protocol but also data analysis and discussion - at least indirectly)

We have taken point 2 into consideration and have brought the theory into the discussion of the findings to line 517 and 601.

  1. Although there are some editorial flaws (e.g. line 20 and 173, 643), these do not affect the quality of the paper and may be corrected at further steps of paper proceeding.

Editorial flaws have been corrected.
